# A Liquid Metal Alternate MHD Disk Generator

Antoine Alemany [1,*], Arturs Brekis [2,3] and Augusto Montisci [4]

1 Centre National de la Recherché Scientifique (CNRS), Institute of Engineering Univ. Grenoble Alpes (Grenoble INP), Laboratoire de Science et Ingénierie des Matériaux et Procédés (SIMAP), University Grenoble Alpes (UGA), 1130, Rue de la Piscine, Domaine Universitaire, BP75, 38402 Saint-Martin, France
2 Institute of Physics, University of Latvia, Miera iela 32, LV-2169 Salaspils, Latvia; arturs.brekis@lu.lv
3 Faculty of Electrical and Environmental Engineering, Riga Technical University, Azenes iela 12/1, LV-1048 Riga, Latvia
4 Electrical and Electronic, English Department, University of Cagliari, Via Marengo 2, 09123 Cagliari, Italy; amontisci@diee.unica.it
* Correspondence: antoine.alemany@grenoble-inp.fr

**Abstract:** In this paper, an electrical generator is presented for the exploitation of alternating energy. Some renewable sources are directly available in such forms, such as the wave power obtainable from the sea, but most of them can be converted to alternative forms; therefore, the proposed generator can be applied to different kinds of renewable sources. In particular, the proposed system is thought to be coupled with a thermoacoustic engine, which converts heat into mechanical vibration without using solid moving parts. This opens the proposed system to the use of most thermal sources, such as solar radiation, waste recovery, geothermic, car exhaust, and others. The object of of this present work concerns the transformation of alternating mechanical energy into electricity by using a specific type of magnetohydrodynamic (MHD) disk generator. The functioning of this generator is based on the interaction between a DC magnetic field embedded in a disk structure and a conducting fluid held in an inner channel. A simplified model of the generator is presented here, and a sensitivity analysis is performed. It is shown that, under specific operating conditions, the efficiency of the system can reach 70% with a level of power of hundreds of watts.

**Keywords:** magnetohydrodynamic disk generator; vibration power; liquid metals; permanent magnets

## 1. Introduction

The main objective of this study is the production of electrical energy in quasi-static conditions. Quasi-static means without any solid moving part, using exclusively the oscillation of a conducting fluid, for example, a liquid metal, to produce electricity. The vibration power that supplies the generator [1] can be provided by a large number of different processes, deriving from either renewable or non-renewable primary sources, such as heat, wind, sea waves [2] and currents, and rivers. An extensive review of the newest thermoacoustic applications is given in [3].

The system proposed in this present paper exploits the induction phenomenon, which makes it possible to separate the internal part of the generator, where the supply power is converted, from the external part, which is connected to the electrical load. The system is comparable with that described in a recent study under the advisement of the EU program SpaceTRIPS, which aimed to produce electrical energy in space in a quasi-static condition that is evidently important for safety reasons. The concept of SpaceTRIPS and the mathematical description is provided in [4], and the first results of the SpaceTRIPS prototype are presented in [5]. This induction process produces electricity at adjustable values of voltage and intensity and, in addition, does not need the presence of electrodes, as in [6], to collect the electric current. The system presented in this paper is based on the same operating principle, including the use of sodium as liquid metal, but only a

single pressure wave is applied to the surface of the liquid instead of two waves in push-pull. This entails a great advantage both in terms of complexity and size of the device and in terms of regulation during operation because there is no need to synchronize the two applied pressures.

As for the electrical power generation stage, several new articles, such as a paper about ocean wave energy converters and energy harvesting applications [7] authored by Domínguez et al. [8] or the numerical studies by Kobayashi [9], justify the creation of an electric generator with a liquid metal working body, thereby eliminating any solid, mechanically degrading moving parts [10]. In [11], a review of a state-of-the-art MHD power generation is given.

One of the first publications in this field of research was devoted to the production of electrical energy in space (E.R.A.T.O project [12]) under the advice of the CNES (French Space Agency) and the CEA (French atomic energy agency). Typically, the technology associated with MHD power generation makes use of conductive generators, which are characterized by low voltage, high current intensity, and DC current, where the example is the OMACON system [13]. In [14], an underwater magnetohydrodynamic acoustic transducer was presented. Later, in [15], Satyamurthy proposed the basic design of a prototype for a liquid metal magnetohydrodynamic power generator for solar and waste heat, and details were studied in [16]. An alternative solution using electrically conducting gas instead of liquid metal is proposed by Montisci et al. in [17]. Moreover, a new series of numerical and experimental research has been presented recently in [18], where another conduction-type liquid metal MHD generator, coupled with a thermoacoustic motor, is constructed with InGaSn as a working body. In [19], a three-stage thermoacoustic engine implementation into the MHD power generator is described. In [20], a mathematical model for the MHD plasma generator, coupled with thermoacoustics, is provided. The functioning of the device proposed in this present article is an alternative to the conductive generators, as the working principle is based on the induction and the operating fluid can be a single-phase or two-phase flow, such as in the OMACON.

The use of vibration power to supply the electrical generator has several advantages, the main one being that the operating fluid is not involved in a continuous flow, which makes it possible to choose a fluid with proper features, such as conductivity, density, and viscosity. Furthermore, the time-varying regime, and consequently the possibility to leverage on the induction phenomenon, together with the absence of solid moving parts, which is typical of MHD devices, enhance the availability of the proposed systems, limit the maintenance, and increase the lifecycle. This aspect allows for implementing the proposed generation system in areas where severe environmental conditions, and the distance from power infrastructures, make maintenance a critical point for the sustainability. Examples of geographical areas with such connotations are deserts in Africa and Asia, wide rural areas in Africa, Asia, and South America, islands, and so on. In general, these territories are affected by high average solar radiation and frequent winds, while the islands can exploit waves and sea currents. Therefore, the proposed technology could facilitate the process of electrification of off-grid territories. At the same time, the proposed technology can also be applied in infrastructure areas [21] by also resorting to other primary sources, such as non-recyclable waste and residual heat from industrial processes.

The present paper is organized as follows. In Section 2, the analytical model is presented, starting with the description of the functioning principle, then describing the magnetic model, and finally, the fluid dynamic model. In Section 3, a sensitivity analysis of the device outputs with respect to the design parameters is given. Finally, conclusions and remarks are given in Section 4.

## 2. Model of the Reciprocating MHD Disk Generator

### 2.1. Principle of Functioning

Figure 1 summarizes the proposed system. The structure of the device is axisymmetric. An oscillating pressure $P'_1$ is applied on the free surface of a conducting liquid contained

in a central tube of internal radius $r_0$. Let $P_0$ be the average pressure of the system. The conducting liquid is forced to oscillate radially inside the disk channel that connects the central column with the toroidal cavity. The cavity is partially filled with the liquid, the remaining volume being filled with a gas of proper features. The free surface of the liquid in the central column and in the cavity oscillates with the same frequency, but the amplitude of the velocity is much higher in the column due to the smaller cross-section. The profile of the channel is assumed to maintain constant the total cross-section along the radius. The oscillating liquid interacts with the applied magnetic field imposed by means of a permanent magnet, which generates a magnetic field of increasing intensity $B_o(r)$ along the radius. The structure of the device is in ferromagnetic material in order to maximize the induction in the volume of the channel. The oscillation of the conducting liquid gives rise to a toroidal-induced current in the liquid itself, which performs as the primary winding of a transformer. A copper coil co-axial with the central column is wrapped in correspondence with the external end of the channel. This coil performs as the secondary winding of a transformer and is electrically connected to the load. As indicated in the Introduction, thanks to inductive coupling, the electrical power is generated without a direct connection between the internal and external parts of the device.

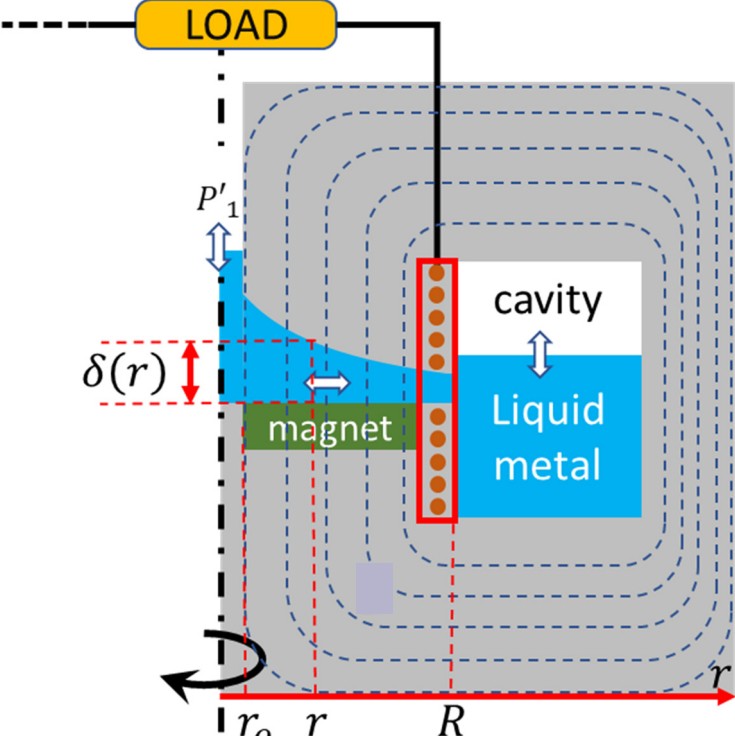

**Figure 1.** Schematic representation of the generator.

Concerning the choice of liquid metal, it can be stated that there are three requests that should be accomplished: low melting point, high electrical conductivity, and low density, to reduce the inertia effects in an oscillating regime. Sodium seems to be an optimal candidate, but it is not the only one. For example, the sodium–potassium alloy (NaK) has more or less the same density and conductivity as pure sodium but has the advantage of having a melting point under 0 °C. Gallium could also be a possible choice, even if its density is a little too high. Electrolytes could represent a valid alternative, even if their conductivity is much lower than that of one of the liquid metals.

Unlike the SpaceTRIPS EU project, which uses more or less the same principles, this engine is mainly intended to work vertically on the Earth to reduce considerably the problem of interface control, even if the instability of the two free surfaces, in the central tube and in the cavity, must be carefully evaluated.

### 2.2. Magnetic Model

The methodology of resolution goes on as follows:

i   The induction equation is solved under the hypothesis of a low magnetic Reynolds number, in which the induced field is much lower than the applied one.

ii  The equation is solved by resorting to Bessel's function. The symmetry condition guarantees that the solution depends on a single complex function $\gamma_1$, which depends on the current in the load. In order to determine this constant, Ampère's theorem is used.

iii Solving the equation of the load circuit gives another relation between the current in the load and the complex function $\gamma_1$.

iv  By combining the results obtained in (ii) and (iii), the complex function $\gamma_1$ is deduced. The current in the load can be expressed in terms of $\gamma_1$.

v   By using the kinetic energy theorem, the pressure at the free surface of the central tube is deduced for a given value of the velocity in the disk channel.

vi  The mechanical power is determined from the scalar product between velocity and pressure, while the electrical power is calculated on the basis of the electrical current circulating in the coil and the electrical load. Finally, the efficiency is calculated as the ratio between the electrical and the mechanical power.

Let us start with the induction equation:

$$\frac{\partial\, b(r)}{\partial t}\vec{k} = \{\nabla \times [\boldsymbol{V} \times (\boldsymbol{B_o}(r,z) + \boldsymbol{b}(r))]\} + \frac{\nabla^2(B_o(r,z) + b(r))}{\mu\,\sigma}\vec{k} \tag{1}$$

where $B_o(r,z)$ represents the DC applied magnetic fieldO and $\boldsymbol{b}(r)$ the induced field, with both essentially vertically directed if the depth $\delta$ of the channel is small compared to its length (aspect ratio $\delta/R \ll 1$). Furthermore, $\boldsymbol{V}$ represents the alternate velocity of the conducting liquid assumed constant along the channel, as the cross-section of the channel is assumed constant along the radius, and $\mu$ and $\sigma$, respectively, represent the magnetic permeability and the electrical conductivity of the liquid metal. Finally, $1/\mu\sigma$ is the magnetic diffusivity of the liquid.

The system is assumed to be axisymmetric so that the derivation in the azimuthal direction vanishes. On the other hand, to keep the velocity $V$ constant along the channel in the radial direction, calling $\delta(r)$ the depth of the channel, it must be

$$V\, 2\pi r\, \delta(r) \equiv constant = V\, 2\pi r_0\, \delta(r_0) \Longrightarrow \delta(r) = \delta(r_0)\frac{r_o}{r} \tag{2}$$

Consequently, $\delta(r)$ decreases as $r$ increases. When the system is at rest,

$$0 = \nabla^2(B_0(r,z)) \tag{3}$$

therefore, the applied magnetic field $B_0(r,z)$ can be eliminated from the diffusion term in (1). The permanent magnets will be designed in such a way that its vertical component $B_o(r)$ increases linearly along the channel:

$$B_o(r) = B_o(r_0)\frac{\delta(r_0)}{\delta(r)} = B_o(r_0)\frac{r}{r_0} \tag{4}$$

In the following, $B_o(r_0)$ will be called simply $B_{ro}$.

On the other hand, assuming the aspect ratio $\delta(r)/r \ll 1$ allows us to neglect the radial component of the induced field. By assuming the hypothesis of small values of the magnetic Reynolds number and considering that the pertinent scale for the induced field is $\delta(r_0)$:

$$Rm\prime = V_o\, \delta(r_0)\, \mu\, \sigma < 1 \tag{5}$$

which implies that the field $b$ is small in comparison with the applied one: $b/B_o(r) \ll 1$. Equation (1) writes

$$\frac{\partial b}{\partial t} = -\frac{1}{r}\frac{\partial}{\partial r}[r\,V\,B_o(r)] + \frac{1}{\mu\sigma}\frac{1}{r}\frac{\partial}{\partial r}\left(r\frac{\partial b}{\partial r}\right) \tag{6}$$

where $V = V_o\,e^{i\omega t}$ is the time-varying velocity of the liquid metal in the channel, and $V_o$ is real and constant along the channel. The induced field $b$ can also be assumed sinusoidal, but unlike the velocity, its amplitude and phase shift vary along the channel: $b = b(r)\,e^{i\omega t}$.

The above assumptions imply that the source term $-\frac{1}{r}\frac{\partial}{\partial r}[r\,V_o\,B_o(r)]$ is constant all along the channel with a value $c = -2V_o\,B_o/r_o$. Only the steady state evolution is considered.

Using the dimensionless variables:

$$r' = \frac{r}{r_o}; \quad b'(r) = \frac{b(r)}{B_o(r_0)}; \tag{7}$$

Equation (6) writes

$$i\frac{Rm\,\omega\,r_0}{V_o}b' = -2\,Rm + \frac{1}{r'}\frac{\partial}{\partial r'}\left(r'\frac{\partial b'}{\partial r'}\right) \tag{8}$$

where $Rm = V_o\,r_o\,\mu\,\sigma$ is taken here with typical scale $r_o$. Equation (8) is a second-order linear ordinary equation with variable coefficients, which has no solution in closed form. It can be solved using Bessel's functions. By removing, for simplicity, the primes in the variables, Equation (8) admits a general solution in the form:

$$b(r) = \gamma_1\,I_o(X) + \gamma_2\,K_o(X) + \frac{2\,i\,V_o}{\omega\,r_0} = \gamma_1\,I_o(X) + \gamma_2\,K_o(X) + \frac{2\,i\,Rm}{Rm^*} \tag{9}$$

where $X = \sqrt{Rm^*\,i}\;r/r_o$, $I_o(X)$, and $K_o(X)$ are the modified Bessel's functions of 0-th order, and $Rm^* = r_0^2\,\omega\,\mu\,\sigma$, which has the same form of the magnetic Reynolds number, characterizes the skin effect. In order to consider $b$ distributed throughout the entire domain, the skin effect must be negligible, and consequently $Rm^*$ must be sufficiently small. This, for given geometry and liquid metal, sets a limit to the angular frequency $\omega$ of the applied pressure $P_1$:

$$\left[\delta(r_o)\cdot\frac{r_o}{r}\right]^2 \omega\,\mu\,\sigma \ll 1 \implies Rm^* = r_0^2\,\omega\,\mu\,\sigma \ll \left[\frac{r}{\delta(r_o)}\right]^2 \tag{10}$$

By assuming the axis of symmetry of the system as a reference, the complex constant $\gamma_2$ vanishes, and then the problem is now reduced to the complex constant of the problem $\gamma_1$.

### 2.3. One-Dimensional Solution

Let us consider Equation (9), taking into account that $\gamma_2 = 0$.

It is assumed hereinafter that all variables, namely the oscillating pressure $p = p_0 e^{i\omega t}$, the current in the load $I = I_0 e^{i\omega t}$, and the current density in the liquid metal $J = J_0 e^{i\omega t}$, have the same frequency $\omega$ and different phase shift, the velocity being assumed as phases origin. The distribution of the induced current inside the liquid metal and then the induced magnetic field depend on the current in the coil. This one can be determined via Ampère's theorem applied to contour 1 and 2 in Figure 2. Referring to contour 1, Ampère's theorem gives that the induced field at the right of the coil can be neglected if the coil is not too far from the end of the channel. Consequently, $b(R + dr) = 0$, and its results, apply Ampère theorem to contour 2:

$$B_{ro}\{b(r_o)\cdot\delta(r_o)\} = \mu n\,I + \mu\,\delta(r_o)\,r_o\int_1^{\frac{R}{r_o}} J(r)r\,dr \tag{11}$$

where $n$ is the number of turns of the coil traveled by an electric current $I$. The induced current density in the liquid metal $J(r) = -(1/\mu)\,(B_{ro}/r_o)\,(db/dr)$ can be deduced from

the evolution of the induced field along the channel. In the above equations, $b$ and $r$ are dimensionless. According to such assumptions, Equation (11) writes:

$$B_{ro}[b(r_o) \cdot \delta(r_o)] = \mu n\, I - B_{ro}\, \delta(r_0) \int_1^{\frac{R}{r_o}} \frac{db}{dr} \frac{1}{r}\, dr \tag{12}$$

and then

$$\gamma_1 B_{ro} \left\{ \delta(r_0) \int_1^{\frac{R}{r_0}} \frac{d[I_0(X)\,]}{dr} \frac{1}{r}\, dr + \left( I_0(X_{r_0}) + \frac{2i\, V_o}{\gamma_1 \omega\, r_0} \right) \delta(r_o) \right\} = \mu n\, I \tag{13}$$

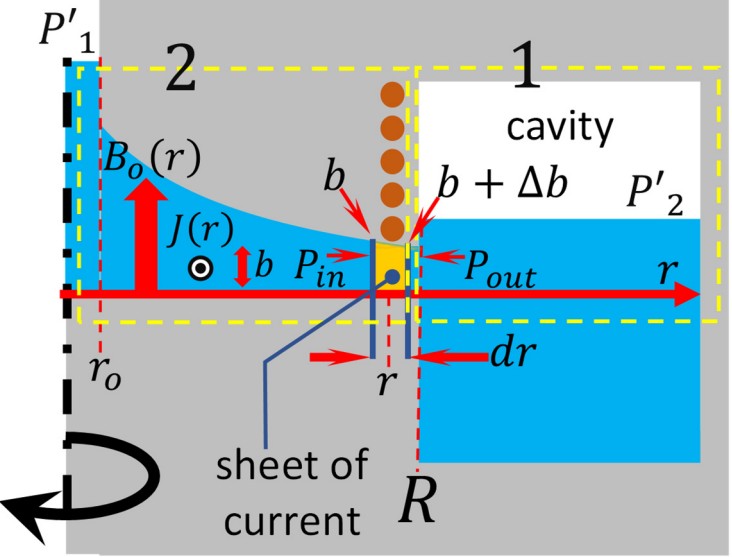

**Figure 2.** Ampère theorem applied to the channel.

Equation (13) is not sufficient to characterize $\gamma_1$ because of the new unknown $I$, namely the current in the coil. Therefore, to solve the system, a new equation is necessary. To this end, the load resistance $\Re$ is involved by considering the load circuit in which the pulsating magnetic flux $\Phi$ across the coil is the source term:

$$-i\, \omega\, \Phi = \Re I \tag{14}$$

The flux $\Phi$ can be expressed in terms of the induction field $b(r)$:

$$-i\, \omega\, n\, B_{ro} r_0^2 \int_1^{\frac{R}{r_0}} b(r) 2\pi r\, dr = \Re I \rightarrow$$

$$-i\, \omega\, n B_{ro} \left[ \gamma_1 r_0^2 \int_1^{\frac{R}{r_0}} I_0(X)\, 2\pi r\, dr + \frac{2i\, V_o}{\omega\, r_0} \left( \pi R^2 - \pi r_0^2 \right) \right] = \Re I \rightarrow$$

$$I = \frac{\omega\, n\, B_{ro}\, r_0^2 \left[ -i\, \gamma_1 \int_1^{\frac{R}{r_0}} I_0(X)\, 2\pi r\, dr + \frac{2\, \pi\, V_o}{\omega\, r_0} \left( \left( \frac{R}{r_0} \right)^2 - 1 \right) \right]}{\Re} \tag{15}$$

By substituting (15) in (14), the value of $\gamma_1$ can be found. To simplify the writing of $\gamma_1$, an approximated value of Bessel's function [22] $I_0$ is assumed:

$$I_0(X) \cong \frac{1}{\sqrt{2\, \pi\, X}}\, e^X = \frac{1}{\sqrt{2\, \pi \sqrt{Rm^*\, i}\, \frac{r}{r_o}}}\, e^{\sqrt{Rm^*\, i}\, \frac{r}{r_o}} \tag{16}$$

where the following change of variables is made:

$$X = \sqrt{Rm^*}\, i\, \frac{r}{r_o} \tag{17}$$

Figure 3 shows the comparison between the values of $I_0$ and the approximated expression (16). The difference tends to zero for high values of $X$, but it is acceptable from $|X| \geq 0.5$. On the other hand, the value of $I_0$ is used essentially by way of integrals that reduces again the level of error.

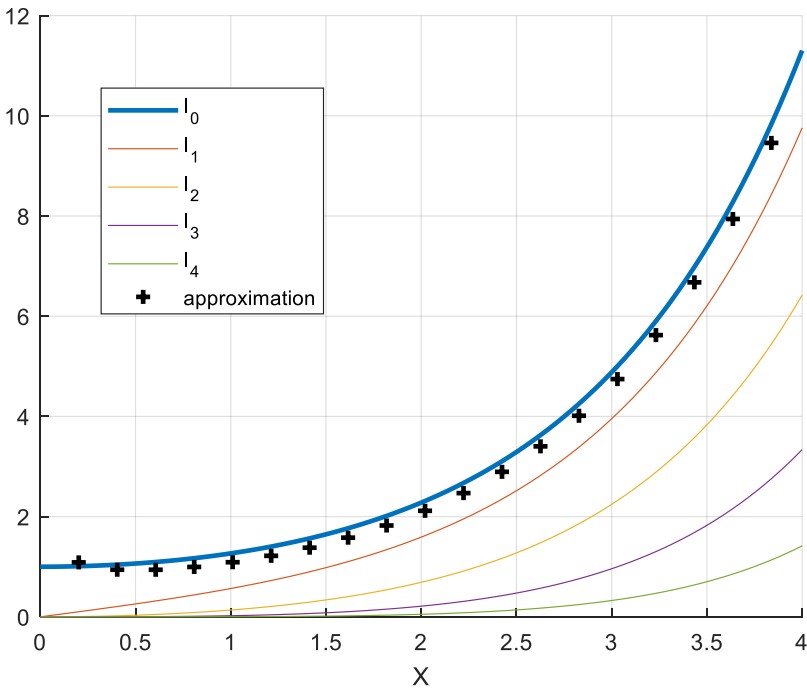

**Figure 3.** Comparison between $I_0$ and the approached Expression (16) for $X$ real.

By using the Expression (16) of Bessel's function and eliminating $I$ from the Equations (13) and (15), $\gamma_1$ can be deduced, and from it, all the other quantities.

$$\gamma_1 = \frac{e^{i\frac{\pi}{8}}\, \frac{2\,\pi Rm\, n^2}{\mathfrak{Re}_{ro}\,\sigma} \left\{ -\frac{i\mathfrak{Re}_{ro}\,\sigma}{n^2\,\pi\,Rm^*} + \left( \frac{R^2}{r_o^2} - 1 \right) \right\}}{\left[ \frac{e^{\sqrt{Rm^*}}\, e^{i\frac{\pi}{4}}}{\sqrt{2\,\pi\,\sqrt{Rm^*}}} + \sqrt{Rm^*} \int_{\sqrt{Rm^*}}^{\sqrt{Rm^*}\frac{R}{r_0}} \frac{1}{\rho}\frac{d}{d\rho}\left( \frac{1}{\sqrt{2\pi\rho}}\, e^{\rho e^{i\frac{\pi}{4}}} \right) d\rho \right] + n^2 i \frac{\sqrt{2\pi}\int_{\sqrt{Rm^*}}^{\sqrt{Rm^*}\frac{R}{r_o}} \sqrt{\rho}\, e^{\rho\, e^{i\frac{\pi}{4}}}\, d\rho}{\mathfrak{Re}_{ro}\,\sigma}} \tag{18}$$

where $\rho = \sqrt{Rm^*}\, r/r_0$, $\delta_R = \delta(R)$, $\delta_{ro} = \delta(r_o)$.

The current $I$ of Equation (15) is expressed in terms of the new variable $\rho$:

$$I = \frac{n\left[ -\gamma_1 \frac{i\omega B_o \sqrt{2\pi}\, r_0^2 e^{-i\frac{\pi}{8}}}{Rm^*} \int_{\sqrt{Rm^*}}^{\sqrt{Rm^*}\frac{R}{r_o}} \sqrt{\rho}e^{\rho\, e^{i\frac{\pi}{4}}}\, d\rho + 2\pi r_0 u_0 B_o \left( \frac{R^2}{r_0^2} - 1 \right) \right]}{\mathfrak{R}} \tag{19}$$

### 2.4. Fluid Dynamic Model

The characteristics of the engine are a function of the driving parameter $(P_1' - P_2')$. In particular, $P_2'$ depends on many factors, such as $P_1'$, the applied pressure on the free surface of the central tube, the inertia of the conducting fluid, the electromagnetic force applied on this fluid in the converging channel, and the characteristics of the gas in the cavity which counterbalances the applied pressure. From that, the mechanical power can be deduced.

The expression of the driving parameter can be obtained by applying the kinetic energy theorem to the full mass of the conducting fluid between the time $t$ and the time $t + dt$:

$$W = \delta E_K = \int_{V_{M'}} \tfrac{1}{2}\, \rho\, V_0^2(t + dt)\ dv - \int_{V_{M'}} \tfrac{1}{2}\, \rho\, V_0^2(t)\ dv =$$
$$= P_1'\, U_1\, S_1\, dt - P_2'\, U_2\, S_2\, dt \tag{20}$$
$$+ \left( \int_{V_{M'}} (\nabla \times b) \times B_0\, dr \right) S\ V_o(t)\, dt$$

where $W$ is the work, $\delta E_K$ is the variation of kinetic energy, $V_{M'}$ is the volume of liquid, $U_1$ is the velocity of the free surface in the central column with cross-section $S_1$ and subject to the pressure $P_1$, $U_2$ is the velocity of the free surface in the cavity with cross-section $S_2$ and subject to the pressure $P_2$, and finally, $S$ is the cross-section of the channel.

By developing the Expression (20) on the three volumes of liquid, it writes

$$\int_{V1} \rho\ \left( V_1\, \tfrac{dV_1}{dt} \right)_t dt\, dv + \int_V \rho\ \left( V\, \tfrac{dV}{dt} \right)_t dt\, dv$$
$$+ \int_{V2} \rho\ \left( V_2\, \tfrac{dV_2}{dt} \right)_t dt\, dv = P_1'\, V_1\, dt - P_2'\, V_2\, dt + \left( \tfrac{1}{\mu} \int_V rot\, b \wedge B\, dv \right) V dt$$

By considering that the velocity along the channel is constant at a given time and because the integral of $\rho\, dv$ over the volume is nothing but the mass of each of the three parts, one obtains

$$M_1\ \left( V_1\, \tfrac{dV_1}{dt} \right)_t dt + m\ \left( V\, \tfrac{dV}{dt} \right)_t dt + M_2\ \left( V_2\, \tfrac{dV_2}{dt} \right)_t dt$$
$$= P_1'\, V_1 S_1\ dt - P_2'\, V_2\, S_2\, dt + \left( \tfrac{1}{\mu} \int_V rot\, b \wedge B\, dr \right)_t s\, V\, dt$$

and, finally,

$$M_1\ (V_1\, i\, \omega\, V_1\ )_t dt + m\ (V\, i\, \omega\, V\ )_t dt + M_2\ (V_2\, i\, \omega\, V_2\ )_t dt$$
$$= P_1'\, S_1\, V_1\, dt - P_2'\, S_2\, V_2\, dt + \left( \tfrac{1}{\mu} \int_V rot\, b \wedge B\, dv \right)_t V\, dt$$

where $M_1$, m, and $M_2$ are, respectively, the mass of fluid in the central column, in the disk channel, and in the cavity. In this equation, the three first terms represent the inertia forces, the two terms on the right-hand side are the power of the pressure forces applied on the free surfaces of the central column and cavity, and the last term is the power of the electromagnetic forces. The same results can be obtained by the application of Bernoulli's theorem in the non-steady-state form and by taking into account the losses due to the Joule dissipation. After reduction, Equation (20) gives

$$P_1' = P_2' + \frac{i\, \omega\, V_C\, M'}{s} + \frac{B_{ro}^2\, \gamma_1 e^{-i\,\frac{\pi}{8}}}{\sqrt{2\, \pi}\ \mu\ \sqrt{R_m^*}} \int_{\sqrt{Rm^*}}^{\frac{R}{r_o}\sqrt{Rm^*}} \frac{\partial}{\partial \rho} \left( \frac{e^{\rho}\, e^{i\frac{\pi}{4}}}{\sqrt{\rho}} \right) \rho\, d\rho \tag{21}$$

In order to characterize the pressure $P_2'$ inside the cavity, the perfect gas law was assumed under the hypothesis of a small variation of this oscillating pressure:

$$\frac{P_2'}{P_0} \ll 1 \tag{22}$$

where $P_0$ is the mean pressure imposed in the system. After some simple first-order developments and assuming adiabatic evolution, the following expression is found:

$$P_2' = -i\, \frac{\gamma\, Po\, V_C\, s}{\omega\, V_{M'}} \tag{23}$$

where $\gamma = C_p/C_v$ is the ratio of thermal capacities at constant pressure and volume, respectively. Finally,

$$P_1' = \frac{Rm}{Rm*}\left(-i\,\frac{\gamma\,Po\,ro\,s}{V_C} + \frac{i\,Rm*^2}{s\,(\mu\sigma)^2}\,\frac{M\prime}{ro^3}\right)$$
$$+\frac{Bo^2}{\mu\,\sqrt{2\,\pi\,Rm^*}}e^{-i\frac{\pi}{8}}\,\gamma_1 \int_{\sqrt{Rm*}}^{\sqrt{Rm*}\frac{R}{ro}} \frac{\partial\left(\frac{e^\rho\,e^{i\frac{\pi}{4}}}{\sqrt{\rho}}\right)}{\partial\rho}\rho\,d\rho \tag{24}$$

The total mass $M$ of the liquid can be subdivided into three volumes, contained, respectively, in the column of length $L_1$ and cross-surface $S_1$, in the channel of length $L$ and cross-surface $s$, and in the cavity of length $L_2$ and cross-surface $S_2$. In each of such volumes, the liquid has a different velocity due to the different cross-sections. To take into account this aspect, an equivalent mass $M'$ is considered:

$$M' = M\left(1 + \frac{L_1}{L}\,\frac{s}{S_1} + \frac{L_2}{L}\,\frac{s}{S_2}\right) \tag{25}$$

with a unique velocity, where momentum is equal to that of one of the three volumes together. The applied mechanical power is deduced from Equation (24):

$$W_{mech} = P_1'\,U_1\,S_1 = P_1'\,V_0\,S \tag{26}$$

which average in time is

$$\overline{W_{mech}} = \frac{\vec{P_1'}.\vec{Uo}\,s}{2} =$$
$$= Re\left\{\frac{Bo^2\sigma\,\sqrt{\frac{\pi}{2Rm^*}}\,Rm\,\delta_{ro}}{(\mu\,\sigma)^2}e^{-i\frac{\pi}{8}}\,\gamma_1 \int_{\sqrt{Rm*}}^{\sqrt{Rm*}\frac{R}{ro}} \frac{\partial\left(\frac{e^\rho\,e^{i\frac{\pi}{4}}}{\sqrt{\rho}}\right)}{\partial\rho}\rho\,d\rho\right\} \tag{27}$$

At the same time, we are interested in the output electrical power, which expression is $\overline{W}_{elec} = \left|\mathfrak{R}I^2/2\right|$:

$$\overline{W}_{elec} = \left|\frac{n^2 B_{ro}^2 2\pi Rm^2}{2(\mu\sigma)^2\mathfrak{R}}\left[\frac{-\gamma_1 i\,e^{-i\frac{\pi}{8}}}{Rm}\int_{\sqrt{Rm^*}}^{\frac{R}{ro}\sqrt{Rm^*}}\sqrt{\rho}e^{\rho e^{i\frac{\pi}{4}}}\,d\rho + \sqrt{2\pi}\left(\frac{R^2}{r_0^2}-1\right)\right]^2\right| \tag{28}$$

where | | indicates the module. From (27) and (28) the efficiency derives

$$\eta = \frac{W_{elec}}{W_{mech}} \tag{29}$$

Mechanical and electrical power are both proportional to $Rm^2$. Therefore, the efficiency does not depend on this parameter. The load factor, the ratio between the self-inductance of the electrical circuit, proportional to $n^2$, and the global resistance of the load and the liquid, this one being proportional to $\delta_{ro}\sigma$, is defined as

$$F = \frac{n^2}{\mathfrak{R}\delta_{ro}\sigma} \tag{30}$$

### 2.5. Summary of the Analytical Results

The previous results can be summarized as below: Equations (24), (27)–(29)

$$P'_1 = \frac{Rm}{Rm*}\left(-i\;\frac{\gamma\;Po\;ro\;s}{V_C} + \frac{i\;Rm*^2}{s\;(\mu\sigma)^2}\frac{M\prime}{ro^3}\right)$$

$$+\frac{Bo^2}{\mu\;\sqrt{2\;\pi\;Rm^*}}e^{-i\frac{\pi}{8}}\;\gamma_1\int_{\sqrt{Rm*}}^{\sqrt{Rm*}\frac{R}{ro}}\frac{\partial\left(e^{\rho\;e^{i\frac{\pi}{4}}}/\sqrt{\rho}\right)}{\partial\rho}\rho\;d\rho$$

$$\overline{W}_{mech}$$
$$= Re\left\{\frac{Bo^2\sigma\;\sqrt{\frac{\pi}{2Rm*}}\;Rm\;\delta_{ro}}{(\mu\;\sigma)^2}e^{-i\frac{\pi}{8}}\;\gamma_1\int_{\sqrt{Rm*}}^{\sqrt{Rm*}\frac{R}{ro}}\frac{\partial\left(e^{\rho\;e^{i\frac{\pi}{4}}}/\sqrt{\rho}\right)}{\partial\rho}\rho\;d\rho\right\}$$

$$\overline{W}_{elec} = \left|\frac{n^2Bo^2Rm^2}{2\;R\;(\mu\sigma)^2}\left[-(\frac{\gamma_1}{Rm})i\;\sqrt{2\pi}\;e^{-i\frac{\pi}{8}}\int_{\sqrt{Rm^*}}^{\sqrt{Rm^*}\;\frac{R}{ro}}\;\sqrt{\rho}\;e^{\rho\;e^{i\frac{\pi}{4}}}\;d\rho\right.\right.$$
$$\left.\left.+2\;\pi\;\left(\frac{R^2}{r_o^2}-1\right)\right]^2\right|$$

$$\eta = \frac{\overline{W}_{elec}}{\overline{W}_{mech}}$$

Both mechanical and electrical powers are proportional to $Bo^2$ and $Rm^2$. Therefore, the efficiency is independent of these parameters. The results depend on the load factor, together with $Rm$, $Rm^*$, and the size of the channel aspect ratio $R/r_o$.

### 3. Results and Discussion

The results are given for $R/r_o = 4$ and $Rm = 3$. As shown in Figure 3, the approximation of the Bessel function is all the more precise when $X$ is large.

Figure 4 shows the dependency of mechanical power with respect to $Rm^*$ and the electrical load. As the independent variable is the velocity of the liquid metal, the mechanical power that has to be supplied, as well as all the other variables, becomes dependent. This trick allows one to greatly simplify the analytical development of the model.

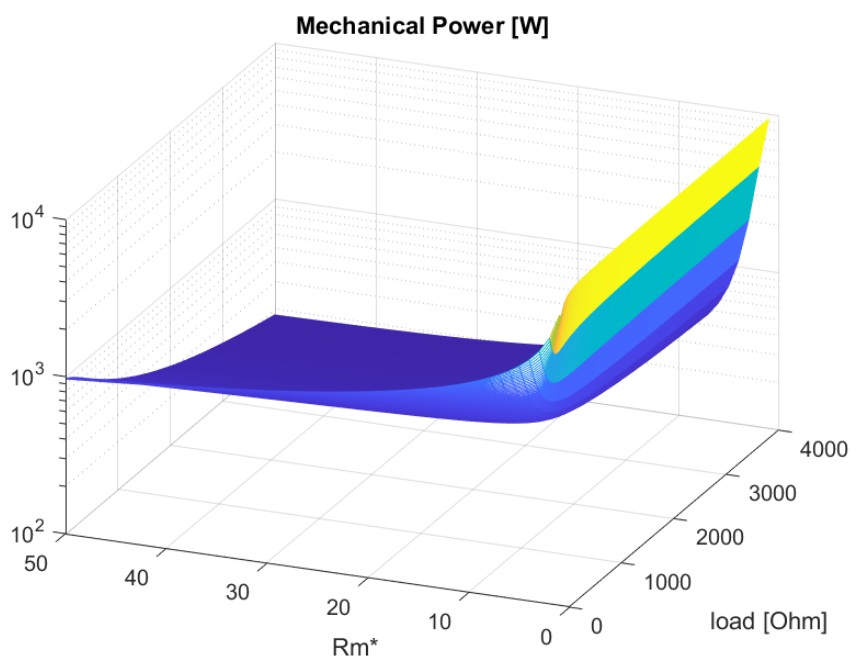

**Figure 4.** Evolution mechanical power versus the load resistance and $Rm^*$, for $Rm = 3$.

The first results concern the trend of electrical power as a function of the load resistance $\mathfrak{R}$ and the screen parameter $Rm^*$. For the large values of $\omega$, which are characterized by high values of $Rm^*$, and as can be seen in Figure 5, the electrical power seems not to be too much affected when $Rm^*$ increases. In Figure 5, the maximum of the power is located around $Rm^* = 20$ and $\mathfrak{R} = 500$ $\Omega$. It has already been noticed that the electrical power is proportional to the squared of $Rm$, namely multiplying $Rm$ by a factor of 2, whereby the electrical power increases by a factor of 4. In Table 1, a possible set of design parameters and outputs are given as an example for a moderate value of electrical power.

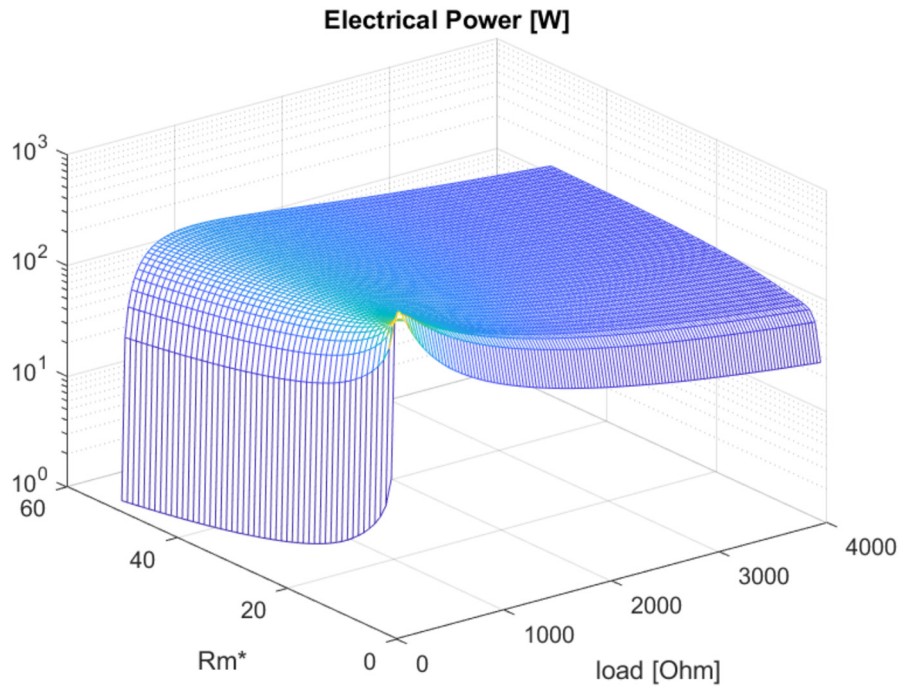

**Figure 5.** Evolution of electrical power versus the load resistance and $Rm^*$, for $Rm = 3$.

**Table 1.** List of the main parameters and possible non-optimized performances of the system.

| | | |
|---|---|---|
| $r_o$ | Internal Radius of the Channel | 40 [mm] |
| $\delta_{ro}$ | Inlet height of the channel | 3 [mm] |
| $R$ | External radius of the channel | 160 [mm] |
| $B_{ro}$ | Applied induction field | 0.3 [T] |
| $\sigma$ | Conductivity of fluid | $10^7$ [S] |
| $\mu$ | Magnetic permeability of the fluid | $4\pi \cdot 10^{-7}$ [H/μ] |
| $n$ | Number of turns of the coil | 280 |
| $V_c$ | Volume of the cavity | $3 \cdot 10^{-2}$ [m$^3$] |
| $M'$ | Equivalent mass of liquid | 0.25 [kg] |
| $s$ | Channel cross-section | 10.05 [cm$^2$] |
| $Rm$ | Magnetic Reynolds number | 3 |
| $Rm^*$ | Skin effect parameter | 20 |
| $Pe_{MAX}$ | Max electric power | 350 [W] |
| $Pm_{MAX}$ | Max mechanical power | 600 [W] |
| $\eta_{MAX}$ | Max efficiency | 55% |
| $P_1$ | Amplitude of pressure | 10 [bar] |

The values in Table 1 are given by assuming sodium as the liquid metal. The obtained values of electrical and mechanical power are sensitive to these parameters. The performance is also dependent on the dimensions of the generator, in particular, on the aspect ratio $R/r_o$. The large number of design parameters and the related sensitivity of performance entails a good strategy to optimize the design of a real device. It can also be remarked that for a given value of $Rm^*$, a maximum of efficiency can be reached versus the evolution of the load factor $n^2/\Re\delta_{ro}\sigma$.

The mechanical power (Figure 4) supplied on the free surface of the central column depends on the phase shift between the velocity and the pressure. The velocity of the liquid in the channel is the reference for the phases; consequently, the pressure is a complex number that depends on the global characteristics of the system and, in particular, on the frequency (through $Rm^*$) and the load resistance (through the load factor). It is important to note that the mechanical power decreases when $Rm^*$ increases, as when $\omega$ increases, the induced current in the channel decreases, thus reducing the electromagnetic forces, which controls the mechanical power.

The electrical power (Figure 5) is null when the resistance is null, then increases rapidly with the resistance to reach a maximum, and then decreases slowly. On the other hand, for a given value of the load resistance, when $Rm^*$ increases ($\omega$ increases), the electrical power seems only slightly affected. This could be interpreted by the fact that the voltage induced in the coil is proportional to the product ($b\cdot\omega$) of the induced field and the frequency. So, even if the induced field decreases, this is compensated by the increase of $\omega$, which finally limits the variation of the voltage.

In Figure 6, the global efficiency is reported. It can be observed that efficiency can reach a value of around 0.7 (in the range of parameters exploited), depending on the value of the load resistance. As expected, when the load resistance is null, the electrical power vanishes, and then the efficiency is null. At the same time, when the load resistance tends to infinity, both the electric and mechanical power do not vary greatly, and the efficiency seems to increase moderately. Consequently, the maximum value of efficiency does not represent, in general, a feasible design, which should be established according to the requirements of each specific application.

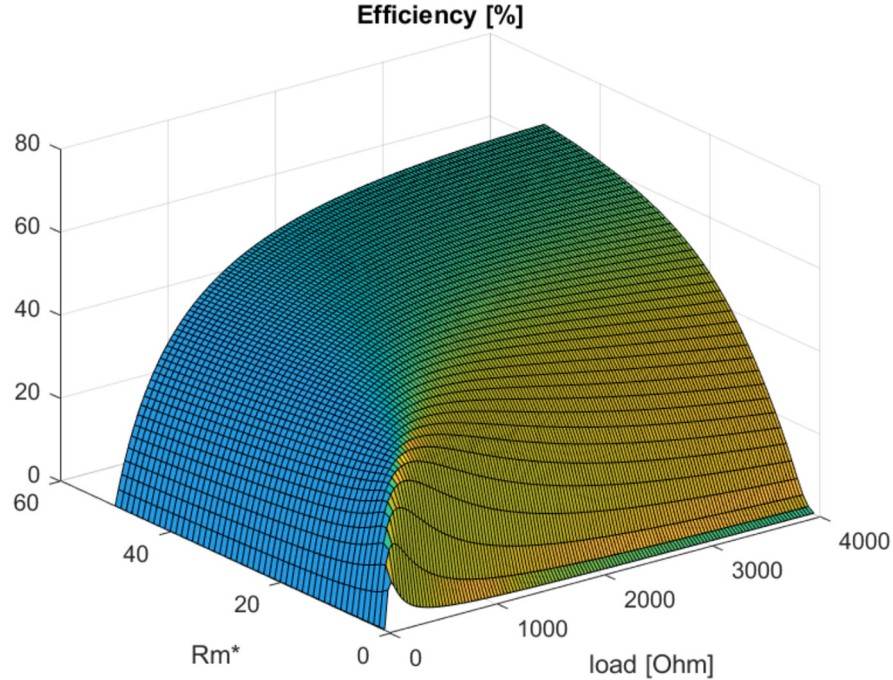

**Figure 6.** Evolution of the efficiency versus the load resistance and $Rm^*$, for $Rm = 3$.

As can be seen in Figure 7, the evolution of pressure presents two characteristic zones. In the first step, a decrease can be observed when $Rm^*$ increases, then it reaches a minimum which depends on the load resistance. With further increasing $Rm^*$, the pressure also increases. These results can be easily interpreted. The amplitude of the pressure can be calculated as the module of Equation (24):

$$
\lfloor P_1' \rfloor = \left| \begin{array}{c} \frac{Rm}{Rm*}\left(-i\ \frac{\gamma\ Po\ ro\ s}{V_C} + \frac{i\ Rm*^2\ M'}{s\ (\mu\sigma)^2\ ro3}\right) + \\[2mm] \frac{Bo^2}{\mu\ \sqrt{2}\ \pi\ Rm^*}e^{-i\frac{\pi}{8}}\ \gamma_1\int_{\sqrt{Rm*}}^{\sqrt{Rm*}\frac{R}{ro}} \frac{\partial\left(e^\rho\ e^{i\frac{\pi}{4}}/\sqrt{\rho}\right)}{\partial\rho}\rho\ d\rho \end{array} \right| \tag{31}
$$

**Pressure [bar]**

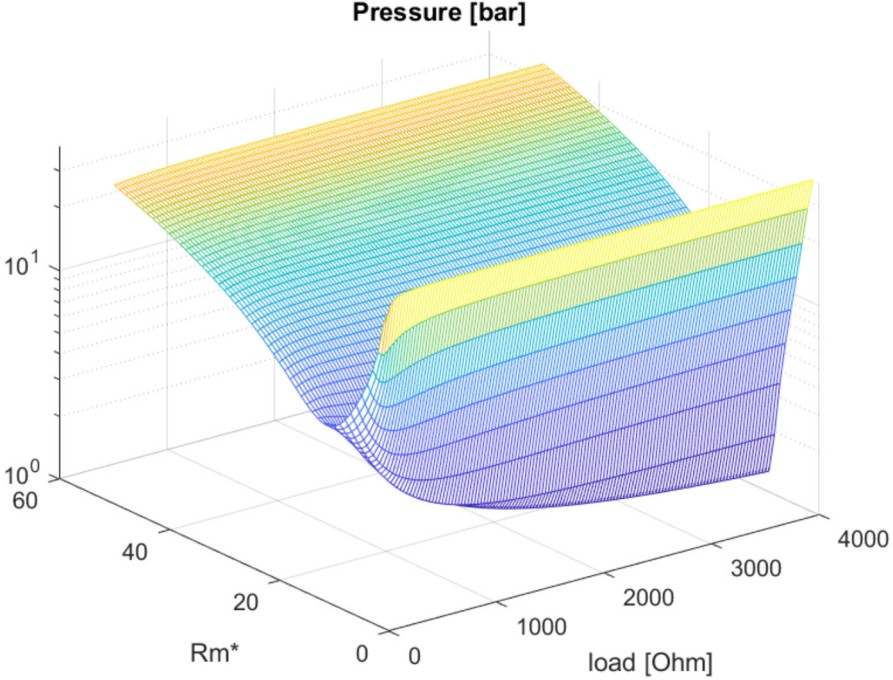

**Figure 7.** Amplitude of pressure to be applied versus the load resistance and $Rm^*$, for $Rm = 3$.

The first term on the first bracket, which corresponds to the counterbalanced pressure in the cavity, is high for low $Rm^*$, compared with the second term in the same bracket, which is due to the influence of the mass of liquid metal. Consequently, the oscillating pressure decreases when $Rm^*$ increases, but when $Rm^*$ becomes sufficiently high, the second term equilibrates the first one. When this happens, the minimum value of the pressure module is reached.

The value of the minimum is given by the term in the second bracket that imposes the mechanical power evolution. As can be seen in Figure 7, the minimum value decreases when the load resistance increases.

The evolution of the induced field along the channel, Figure 8, presents maximum values near the outlet; it is almost null at the inlet and on a large range of the radius. On the other hand, considering that the evolution of the applied field increases along the radius, that gives for $r = 14\ cm$, $B_{(r=0.14)} = 1.05$ [T], the local values of the ratio $b$(induced)$/B$ (applied) are, respectively,

$$\sim 0.5 \text{ for } Rm^* = 10$$
$$\sim 0.2 \text{ for } Rm^* = 20$$
$$\sim 0.16 \text{ for } Rm^* = 30$$

and is not negligible for $Rm^* = 10$, but it is in agreement with the assumption of a small magnetic Reynolds number. The change of sign at the end of the channel is due to the influence of the coil, which, according to the Lentz law, is opposite to the cause which

generated it, which is the induced current in the channel. On the other hand, after the coil location, here at 0.16 [m] from the origin, the induced field vanishes.

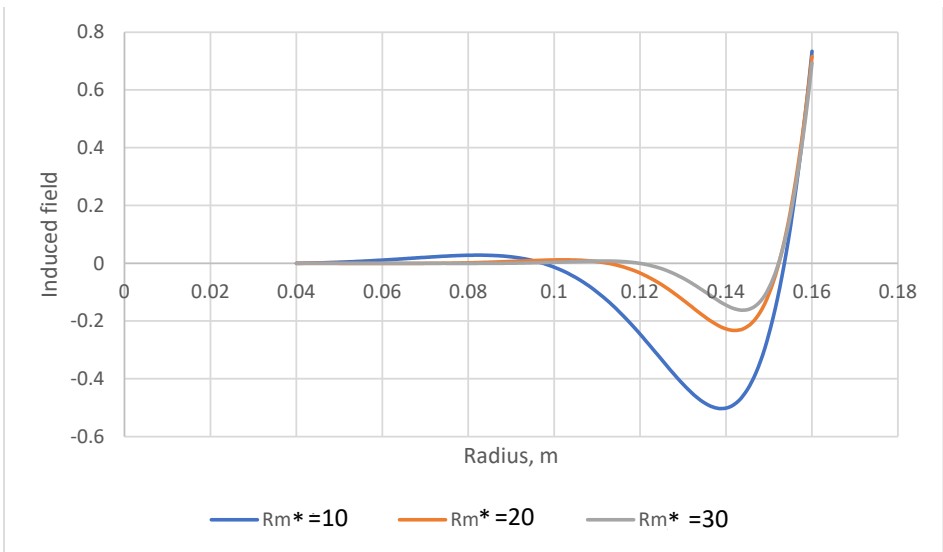

**Figure 8.** Induced field along the radius for a load resistance of 2130 [Ω], for $Rm = 3$ and aspect ratio $R/r_o = 4$.

It can also be noticed that the induced field is proportional to $Rm$ and decreases when $Rm^*$ increases, in agreement in particular with Equation (9). It is an easy task to extrapolate its intensity for any values of magnetic Reynolds number from Figure 8.

In Figure 9, for a fixed value of $Rm = 3$ and $Rm^* = 20$, the electrical power trend is reported depending on the load resistance.

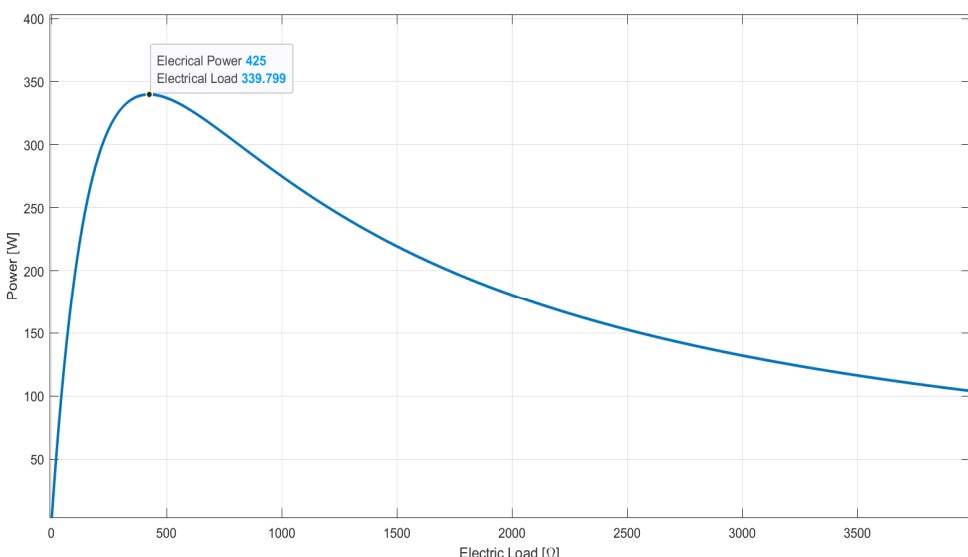

**Figure 9.** Electrical power for $Rm = 3$, $Rm^* = 20$, vs. the load resistance.

On the other hand, to ensure the penetration of the pulsating-induced field in the cross-section of the channel, the value of $Rm^*$ must not be too high. In practice, the following recommendation should be satisfied:

$$0.2 \lesssim Rm* \lesssim \left(\frac{r_o}{\delta_{ro}}\right)^2 \backsim 100 \tag{32}$$

Taking into account the diffusivity of liquid metal such as $\mu_m = 1/\mu\sigma$ and imposing that the time of penetration of the induced field must be lower than the typical time of evolution of the system of order, $1/\omega$ requires $Rm^*$ to be lower than 100. On the other hand, $Rm^*$ must be higher than 0.2 to accept the approximation $I_0$.

By considering now the scale parameters, it can be remarked that the trend of the performance can be deduced from the following laws:

- The trends of both electrical and mechanical powers vary with $Rm^2$ and are both proportional to $B_0^2$.
- It results that the efficiency is independent of both the magnetic Reynolds number and the applied magnetic field.

These considerations are valid in the frame of the assumed assumptions and, in particular, neglecting the viscosity. This hypothesis could be accepted if the boundary layer along the wall of the channel is everywhere much lower than the depth of the channel. This could be estimated by considering the following parameter:

$$\frac{\delta_\nu}{\delta_R} \sim \sqrt{\frac{1}{R_\omega}} \ll 1 \rightarrow R_\omega \gg 1 \tag{33}$$

where $R_\omega = \omega\, \delta_R^2/\nu$ is the Reynolds number with typical velocity $\omega\, \delta_R$, the typical length is $\delta_R$, and $\nu$ is the kinematic viscosity. In practice, this condition is well satisfied for $\omega > 10\ [\text{rad}/\text{s}]$, which is a very low frequency.

Let us now consider the adaptation of the generator with the system producing mechanical work, for example, a thermoacoustic engine. Such a system has its own performance in terms of velocity, pressure, and phase shift between these two entities. So, the relation between applied pressure $P'_{ap}$ and applied flow rate $Q_{ap}$ can be characterized by the impedance $Z_{ap}$, such as

$$P'_{ap} = Z_{ap}\, Q_{ap} \rightarrow Z_{ap} = \frac{P'_{ap}}{Q_{ap}} \tag{34}$$

Everything being fixed, $Z_{ap}$ is a function of the angular frequency $\omega$. On the other hand, the prime mover needs to work with an equivalent impedance, such as

$$P'_1 = Z_{gen}\, Q_{gen} \rightarrow Z_{gen} = \frac{P'_1}{Q_{gen}} \tag{35}$$

which is also a function of $\omega$. So, the identity $Z_{ap} = Z_{gen}$ fixes the frequency, the working point, and the main characteristics of the two coupled engines. The best value in terms of efficiency for a given geometry can be adjusted by way of the load resistance.

By considering that the velocity is the reference for the phase shift, the impedance of the disk generator is proportional to $P'_1$. After some calculation, the impedance of the generator is

$$Z_{gen} = \frac{P'_1\cdot}{Q_0} = \frac{P'_1}{2\,\pi\,r_o\delta_{ro}\,V_o} \tag{36}$$

The optimality of the working point is affected by the working conditions, for example, by the change of the heat source which supplies the driving machine. In this case, a new optimal point must be found, which can be obtained by changing the value of the load resistance.

## 4. Conclusions and Final Remarks

The electrical disk generator presented in this paper can be connected with several systems producing alternate mechanical power. The coupling between the prime mover and electrical generators must be realized by the way that the two impedances coincide.

One of the possibilities of using the electrical disk generator is founded on the use of a thermoacoustic engine [1]. Such coupling is represented in Figure 10. The thermoacoustic engine produces a traveling wave pressure in a loop when it is submitted to a temperature gradient greater than a critical value. The pressure wave is applied on the free surface of the disk electrical generator, supplying the vibration on the conducting liquid in the channel of the disc generator and so producing electricity as described in this present paper.

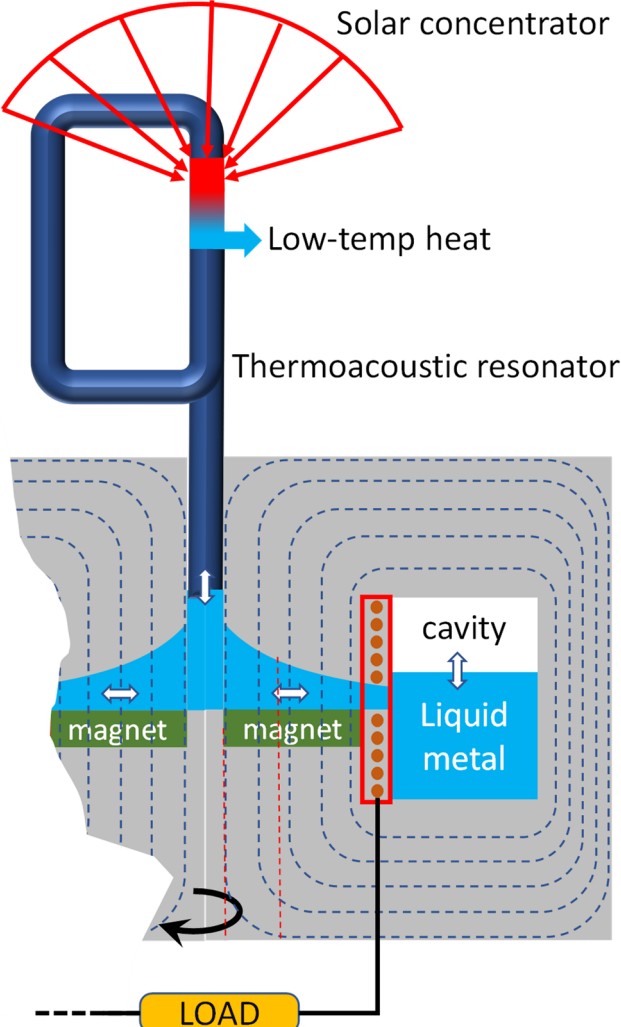

**Figure 10.** Principle of a thermoacoustic solar engine equipped with an electrical disk generator.

Solar radiation is a particularly suitable source of thermal energy by means of a parabolic concentrator. It is worth noting that the described layout does not entail any solid moving part, which implies several advantages in terms of efficiency, cost of materials, power density, availability, lifecycle, and maintenance [2,6,8]. It is consequently well adapted for applications in off-grid areas, especially in low-income territories, or in space applications, where volumes, mass, and inertia play a critical role.

The efficiency of the system can reach 50% of the Carnot efficiency, which means that the absolute efficiency is related to the level of temperature that is reached in the focus of the concentrator [23]. Open issues are the thermal insulation of the thermoacoustic loop and the need for high pressures (around 40 bars), as the oscillating pressure and then the level of power should range from 5 to 10% of the mean pressure. Other suitable renewable sources of heat are exhausts of industrial processes and geothermal energy. It has been demonstrated that an interval of 100 [K] is sufficient to trigger the thermoacoustic effect, even if the efficiency is very low. It is worth noting that the system is suitable for the

exploitation of heat at high temperatures, and it conjugates high efficiency with intrinsic safeness, as it involves the flow of energy without a corresponding flow of matter. This makes the system particularly indicated for nuclear plants. As a final remark, it is worth pointing out that the disk generator presented in this paper is supplied by an oscillating pressure; therefore, any primary source that can be converted into mechanical vibration is a valid candidate as a driver of the generator. Examples of non-thermal primary sources which fall into this category are sea waves, sea currents, and wind.

In the future, the coupling of the disk generator presented in this paper with the mentioned primary sources will be considered.

**Author Contributions:** Conceptualization, A.A.; methodology, A.A.; software, A.B. and A.M.; validation, A.A. and A.M.; formal analysis, A.A., A.B. and A.M.; investigation, A.B.; resources, A.B. and A.M.; data curation, A.B. and A.M.; writing—A.A.; writing—review and editing, A.B. and A.M.; visualization, A.M.; supervision, A.A. All authors have read and agreed to the published version of the manuscript.

**Funding:** This research received no external funding.

**Institutional Review Board Statement:** Not applicable.

**Informed Consent Statement:** Not applicable.

**Data Availability Statement:** Data sharing not applicable.

**Conflicts of Interest:** The authors declare no conflict of interest.

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
