# Peer review of "A Liquid Metal Alternate MHD Disk Generator"

_sustainability, doi:10.3390/su151612619_

Round 1

Reviewer 1 Report

The manuscript presents a liquid metal alternate magnetohydrodynamics disk generator to enhance the performance of a thermoacoustic generator. The manuscript needs a major revision to be published in the journal Sustainability.

1) Abstract needs a revision:

i) References should be avoided in the Abstract. Refs [1] and [2] should be mentioned in the Introduction section.

ii) The sentence on lines 17-18 is incomprehensible and should be revised.

iii) The main findings of the study should be presented in the Abstract.

2) It is stated in the Introduction section that “The system is comparable with that described in a recent study under the advisement of the EU program spaceTRIP [8], [9] …” and “The functioning of the proposed device is alternative to the conductive generators, …”. However, the novelty of the proposed device is not mentioned in the Introduction. The authors should present the novelty of the device and its contribution to the literature.

3) The manipulations that yield Eq. (13) should be presented.

4) Subscripts are written in an incorrect manner in some of the equations. For instance, Eqs. (15) and (20) should be revised. The authors should check all equations.

5) What does the imaginary part of the term in Eq. (16) correspond to? An explanation is required.

6) Section 3 should be entitled as “Results and discussion”.

7) Figure 4 should be mentioned prior to Figure 5.

8) In the caption of Table 1 it is stated that “This table as all the other figures are given with the hypothesis than the liquid metal is the sodium.” The information regarding the liquid metal is important and should be presented in the text. It should be given in both section 1 and section 3.

9) There are some minor typographical errors in the manuscript. For instance, the question mark on line 298 should be deleted, the wording “filed” on line 342 should be revised and “Figure 11” on line 405 should be corrected as “Figure 10”. The authors should check for other errors as well.

10) On line 347 it is stated that “The following figures propose a more easily reading of the results.” The referent figures should be clearly expressed.

11) A reference should be provided for the recommendation given on line 358.

There are some minor typographical errors in the manuscript. The authors should check the paper.

Author Response

The authors wish to thank the reviewers for their accurate check of the manuscript and for the fruitful suggestion, which allowed to significantly improve the quality of the paper. In the following pages the answers to the remarks are reported.

Reviewer 2 Report

The quality of the paper needs to be improved.

Additionally:

- Figures are very poor in quality and difficult to understand.

- Figures 4, 5, 6 and 7 cannot be analyzed since the axes are not conveniently identified. IS units may be used, also.

- In Fig. 8 units are missing.

- In general, equations quality and format should be revised and not all of them are numbered.

Author Response

The authors wish to thank the reviewers for their accurate check of the manuscript and for the fruitful suggestions which  allowed to significantly improve the quality of the paper.

In the following pages the answers to the rematks are reported

Reviewer 3 Report

The system described in this paper is intended to valorise the sources of pulsating energies. This paper is intended to enhance the energy obtained by the thermoacoustic generator able to deliver mechanical energy from different heat sources at adjustable frequencies and powers. These systems convert heat into mechanical energy in the form of an acoustic wave. They are compatible with main of thermal sources, solar radiations, waste recovery, geothermic, car exhaust and others. The object of the present work concerns the transformation of the oscillating mechanical energy into electricity by using a specific type of MHD generator. The functioning of this generator is based on the interaction between a DC magnetic field embedded in a disk structure, with a conducting fluid circulating in the channel of this structure. A simplified model of the generator is presented here, and a sensitivity analysis is performed. There are major amendments which should be incorporated. Please find comments as below:

1. Referencing should be avoided within abstract. The best findings of the study should be provided within abstract.

2. Each reference should be discussed separately within introduction.

3. There is lack of critical analysis within introduction which should be supported using the most important and recent articles in the field.

4. Novelty section should be restructured to expose the importance of this study to the subject in the field. 

5. Methodology should be supported by a comprehensive schematic diagram.

6. Additional reasoning phenomenon should be provided to support the obtained results.

7. Quality of figures should be improved extensively.

8. Comprehensive proofread is essential to rectify the typo/grammatical errors. 

Comprehensive proofread is essential to rectify the typo/grammatical errors. 

Author Response

The authors wish to thank the reviewers for their accurate check of the manuscript and for the fruitful suggestions which allowed to significantly improve the quality of the paper.

In the following apges the answers to the remarks are reported.

Round 2

Reviewer 1 Report

The paper can be published in the journal Sustainability following a minor revision:

1) The revised version of the manuscript contains some grammatical errors. The authors should correct them:

i) Line 14: “… can be converted to alternating …”

ii) Line 18-19: “… such as solar radiation, …”

iii) Line 48: “… principle, including the use …”

iv) Line 57: “In [14] there is a review …”

v) Line 271: “In this equation …”

vi) Line 275: “… and taking into account …”

2) Figures 5-7 are presented as tiny figures in the revised version of the manuscript. These figures should be checked.

The revised version of the manuscript contains some grammatical errors. Please see the comments.

Author Response

Thank you for your comments. The corrections have been made.

Reviewer 2 Report

Figs. 5,6,7 are not correctly displayed

Author Response

Thank you for yours remarks. The figures have been modified.

Reviewer 3 Report

Accept.

Accept.

Author Response

Thanks for your effort.
